# Outer Membrane Vesicles of *Avian Pathogenic*
*Escherichia coli* Mediate the Horizontal Transmission of *bla*_CTX-M-55_

**DOI:** 10.3390/pathogens11040481

**Published:** 2022-04-18

**Authors:** Chao Li, Renqiao Wen, Rongrong Mu, Xuan Chen, Peng Ma, Kui Gu, Zheren Huang, Zijing Ju, Changwei Lei, Yizhi Tang, Hongning Wang

**Affiliations:** 1Animal Disease Prevention and Food Safety Key Laboratory of Sichuan Province, College of Life Sciences, Sichuan University, No. 29 Wangjiang Road, Chengdu 610064, China; lichaomeet@gmail.com (C.L.); wenrenqiao132@163.com (R.W.); 2019222040101@stu.scu.edu.cn (X.C.); ma_peng99@163.com (P.M.); gukui0404@stu.scu.edu.cn (K.G.); jeremy96101817@163.com (Z.H.); jzjscu@163.com (Z.J.); leichangwei@126.com (C.L.); tangyizhi23@163.com (Y.T.); 2Key Laboratory of Bio-Resource and Eco-Environment of Ministry of Education, College of Life Sciences, Sichuan University, No. 29 Wangjiang Road, Chengdu 610064, China; 3Provincial Key Laboratory for Transfusion-Transmitted Infectious Diseases, Institute of Blood Transfusion, Chinese Academy of Medical Sciences and Peking Union Medical College, Chengdu 610052, China; rongrongxmu@163.com

**Keywords:** *avian pathogenic Escherichia coli* (APEC), *bla*
_CTX-M-55_, outer membrane vesicles (OMVs), horizontal gene transfer

## Abstract

The CTX-M-55 type extended-spectrum β-lactamase (ESBL) producing *Enterobacteriaceae* is increasing in prevalence worldwide without the transmission mechanism being fully clarified, which threatens public and livestock health. Outer membrane vesicles (OMVs) have been shown to mediate the gene horizontal transmission in some species. However, whether *bla*_CTX-M-55_ can be transmitted horizontally through OMVs in *avian pathogenic Escherichia coli* (APEC) has not been reported yet. To test this hypothesis, an ESBL-producing APEC was isolated and whole-genome sequencing (WGS) was performed to analyze the location of *bla*_CTX-M-55_. Ultracentrifugation and size exclusion chromatography was used to isolate and purify OMVs, and the transfer experiment of *bla*_CTX-M-55_ via OMVs was performed finally. Our results showed that the *bla*_CTX-M-55_ was located on an IncI2 plasmid. The number and diameter of OMVs secreted by ESBL-producing APEC treated with different antibiotics were significantly varied. The transfer experiment showed that the OMVs could mediate the horizontal transfer of *bla*_CTX-M-55_, and the frequency of gene transfer ranged from 10^−5^ to 10^−6^ CFU/mL with the highest frequency observed in the Enrofloxacin treatment group. These findings contribute to a better understanding of the antibiotics in promoting and disseminating resistance in the poultry industry and support the restrictions on the use of antibiotics in the poultry industry.

## 1. Introduction

*Avian pathogenic Escherichia coli* (APEC) is an important pathogenic bacterium that endangers the poultry industry, and its pathogenic mechanism is still unclear. APEC in chickens, turkeys, and other poultry mainly elicit extra-intestinal diseases such as airsaculitis, polyorrhymenitis, and sepsis [1]. Besides, it has been proposed that APEC may be an important source of human *extra-intestinal pathogenic Escherichia coli* (ExPEC), which shares virulence genes with *uropathogenic Escherichia coli* (UPEC), and *neonatal meningitis causing Escherichia coli* (NMEC), thus threatening human health [2]. The extended-spectrum β-lactamase producing *E. coli* (ESBL-E) poses an enormous challenge to β-lactam antibiotics though it is currently the most commonly used antibiotics in the clinical treatment of Gram-negative bacterial infections. The main resistance mechanism of β-lactam antibiotics is extended-spectrum β-lactamases (ESBLs). Among them, CTX-M-type hydrolase is known as the most popular ESBLs, which undergoes preferential hydrolysis of Cefotaxime. The *bla*_CTX-M_ is often reported in *Enterobacteriaceae* isolated from animals and humans, especially *E. coli* [3]. The ESBL-producing *E. coli* has even been found in environments with little or no human activity, such as the rainforest of Tunisia and non-zed agriculture soils [4]. The gene encoding CTX-M-type hydrolase was originally derived from the chromosome of *Kluyvera*, which was almost nonpathogenic to humans [5]. However, at least 244 genetic subtypes of CTX-M-type hydrolases have so far been identified in the Gram-negative clinical pathogens.

The CTX-M-55 producing *E. coli* was first discovered in Thailand in 2006 [6] and has spread rapidly in dozens of countries around the world [7,8,9,10,11]. The genetic environment analysis of *bla*_CTX-M-55_ showed that it is usually located downstream of the insertion sequence *ISEcp1*, which facilitates its inter-replicon mobility in *E. coli* hosts by plasmids (IncI, IncK, IncFII, IncHI2, etc), indicating that plasmids play an important role in the horizontal transmission of *bla*_CTX-M-55_ [12]. Plasmid-mediated gene horizontal transfer is also the main reason for the wide and rapid spread of *bla*_CTX-M-55_ [3]. In addition to humans, the distribution of *bla*_CTX-M-55_ is also involved in poultry, livestock, water sources, vegetables, wild animals, and plants [13,14], which poses a serious threat to public health. In recent years, *bla*_CTX-M-55_ producing *E. coli* has been found to be globally prevalent, and it was reported that the detection rate of *bla*_CTX-M-55_ producing *E. coli* in China is 18.40% [15].

Outer membrane vesicles (OMVs) are spherical structures composed of bilayered membrane naturally secreted by Gram-negative bacteria with diameters ranging from 20 nm to 400 nm, which contains toxins, peptidases, lipopolysaccharides (LPSs), and nucleic acids [16]. The membrane structure is usually composed of proteins from the periplasm and cell membrane. Membrane vesicles of Gram-positive bacteria also called EVs are produced when a section of the cytoplasmic membrane protrudes and buds off selectively encapsulating various components [17]. As a medium of microbial community interaction, OMVs or EVs can transfer protease, glycosidase, and peptidase encoding genes to the bacteria to cope with different types of environmental stress and which plays an important role in bacteria self-protecting [18].

In recent years, OMVs-mediated horizontal transfer of drug resistance genes has been reported in *Acinetobacter baylyi*, *Escherichia coli* O104: H4, *Streptomyces coelicolor*, *Klebsiella pneumoniae*, and other pathogens. Teresa et al. analyzed the OMVs from different stages of *S. coelicolor* by proteomics and found that the components of OMVs in different stages were complex, including the proteins related to cell metabolism, molecular transport, and stress. The protein composition of OMVs produced by different stages of *S. coelicolor* is significantly different, which also indicates that bacteria may produce OMVs with different biological functions under different growth phases [19]. Although some studies have shown that OMVs are the product of cell death, the obvious difference of OMVs produced under different conditions also further indicates that there is an unclear mechanism of the regulation and control of OMVs formation as well as control of the selectively encapsulating different substances [20]. Shweta et al. purified the OMVs secreted by *A. baylyi* under different antibiotic treatments and found that these OMVs are varied in size, protein, and nucleic acid content. Moreover, OMVs isolated from *A. baylyi* that knocked the *comA* or *comB-comF* out could not mediate the transfer of plasmid, indicating that the OMVs-mediated DNA transfer was regulated by some regulatory mechanisms [21]. Federica et al. calculated the plasmid copies of the OMVs isolated from *K. pneumoniae* and analyzed the transfer frequency of plasmids, found that the OMVs had a protective effect on plasmids and could transfer plasmids effectively [22]. In 2020, Martina et al. published a study on the OMVs-mediated *bla*_CTX-M-15_ horizontal transfer, which found that the OMVs isolated from *Escherichia coli* O104: H4 could mediate the horizontal transfer of *bla*_CTX-M-15_, which located on a 90 kb plasmid and ciprofloxacin can elevate the transfer frequency [23]. The above studies have shown that OMVs are efficient means to mediate the gene horizontal transfer.

The OMVs-mediated gene horizontal transfer is mainly carried out by wrapping the nucleic acid DNA, which helps to protect the inside nucleic acid from external nuclease degradation and can also achieve the purpose of long-distance direct transport. Therefore, OMVs are likely to be efficient tools for communication among microbial communities evolved from bacteria, and thus also named ‘secretion system type zero’ [24]. However, there is still limited research on OMVs-mediated drug resistance gene transmission, and the experimental strains used in each study are varied. It is unclear whether OMVs-mediated horizontal transfer of antibiotic resistance genes is universal or not. The ability of important animal pathogens to produce OMVs under different resistance conditions and the mechanism of OMVs-mediated drug resistance gene transmission are still unclear and remain to be clarified. In addition, exploring new ways of drug resistance gene transmission will also help us to better understand drug resistance and propose reasonable intervention measures. In this study, ESBL-producing APEC was isolated from the sick-laying hen sample. Whole-genome sequencing (WGS) was used to analyze the resistance gene that existed in the ESBL-producing APEC. OMVs released by ESBL-producing APEC were isolated and purified for the detection of the drug resistance gene. Subsequently, the transfer experiment was performed to evaluate the horizontal transmission of *bla*_CTX-M-55_ mediated by OMVs.

## 2. Results

### 2.1. Identification of an IncI2 Plasmid Bearing the bla_CTX-M-55_

An APEC named *E. coli* SCAO22 was isolated and identified from a sick-laying hen sample. Antibiotic susceptibility testing showed that the *E. coli* SCAO22 exhibited resistance to β-lactams including Cefotaxime (CTX), Ceftazidime (CAZ), Amoxicillin (AML), Ampicillin (AMP), Aztreonam (ATM), Cefoxitin (FOX), quinolone antibiotics Enrofloxacin (ENR) and chloromycetin antibiotics Florfenicol (FFC) but sensitive to the carbapenem antibiotic Meropenem (MEM) (Appendix A). The analysis of WGS results showed that the *E. coli* SCAO22 (GenBank accession number JALGQR000000000) possessed *bla*_CTX-M-55_, *bla*_TEM-1B_, *aac(3)-IV*, *aph(3’)-Ia*, *sul2*, *tet(B)*, *catA1*, *floR*, *dfrA17*, *aph(6)-Id*, *aph(6)-Id*, *mph(A)*, *sitABCD*, with 4.96 M genome size, the GC content was 50.5%, the N50 value was 94,783, and the L50 value was 17. The detection of the virulence gene showed that it carried typical APEC virulence genes, including *iroN*, ompT, *iss*, *iutA*, *hlyF*, *IbeA*, *Hcp*. Further genetic environment analysis of the resistance gene *bla*_CTX-M-55_ showed that the *bla*_CTX-M-55_ was located on an IncI2 plasmid. The complete sequence of the plasmid was obtained by comparing it with the NCBI database and PCR splicing (Figure 1). The *bla*_CTX-M-55_ -bearing plasmid was 62,196 bp in size and named pCTX-M-55 (GenBank accession number OL539428).

### 2.2. Extraction and Characteristic Analysis of OMVs

The OMVs were extracted from *E. coli* SCAO22 by ultracentrifugation (UC) and size exclusion method (SEC). Transmission electron microscope (TEM) observation showed that the membrane structure of OMVs was intact and presented the shape of classic saucer-like vesicles. Flow nanoanalyzer (nFCM) measuring showed that the average diameter of OMVs from *E. coli* SCAO22 without antibiotic treatment (hereafter named control group) was 79.42 nm, and the average diameter of OMVs from *E. coli* SCAO22 with 128 μg/mL Amoxicillin treatment (hereafter named Amoxicillin group) was 60.14 nm, the average diameter of OMVs from *E. coli* SCAO22 with 4 μg/mL Enrofloxacin treatment (hereafter named Enrofloxacin group) was 64.18 nm. Compared with the control group, the mean diameter of OMVs generated from antibiotic treatment was smaller. Additionally, the concentration of OMVs generated after antibiotic treatment was increased significantly compared with the control group, in which Enrofloxacin group (5.66 ± 1.2 × 10^12^ particles/mL) > Amoxicillin group (8.89 ± 0.36 × 10^11^ particles/mL) > control group (2.26 ± 0.78 × 10^10^ particles/mL), as shown in Appendix A and Figure 2. Meanwhile, we found that the concentration of nucleic acid and protein in the isolated OMVs after antibiotic treatment was increased (Figure 3a). Western blot (WB) was used to confirm the presence of OmpA and the outer membrane protein A (OmpA) as a marker of *E. coli* OMVs (Figure 3b). β-NADH oxidase activity was not observed in the OMVs preparations, whereas the oxidase activity was found in bacterial lysate due to the presence of inner membrane material, as expected, which indicated that the extracted OMVs were not contaminated with cytoplasmic proteins (Figure 3c).

### 2.3. Detection and Quantitation of bla_CTX-M-55_ in OMVs

The WGS showed that *bla*_CTX-M-55_ was located on an IncI2 plasmid named pCTX-M-55. PCR results showed that all the OMVs originating from either the control group or the Amoxicillin treatment group and the Enrofloxacin treatment group contained the resistance gene of *bla*_CTX-M-55_ (Figure 4a). The absolute qPCR results showed that the copies of *bla*_CTX-M-55_ varied in different OMVs groups, 30,615 copies/μL in the Enrofloxacin treatment group, 1196 copies/μL in the Amoxicillin treatment group, and 208 copies/μL in the control group (Figure 4b).

### 2.4. OMVs from E. coli SCAO22 Can Mediate the Transfer of bla_CTX-M-55_ to E. coli C600

A rifampin resistance *E. coli* C600 (>1024 μg/mL) was used as the recipient strain and co-incubated with OMVs isolated from different SCAO22 (grown without antibiotics (the control group) or with 128 μg/mL Amoxicillin (the Amoxicillin group) or 4 μg/mL Enrofloxacin (the Enrofloxacin group)), respectively. The DNA transformation was firstly ruled out because when the plasmid was incubated with *E. coli* C600, there was no colony grown on the plate with 200 μg/mL rifampin and 4 μg/mL Cefotaxime. Additionally, the group of lysed OMVs had no colony grown on the plate with 200 μg/mL rifampin and 4 μg/mL Cefotaxime while only intact OMVs led to cefotaxime-resistant colony recovery, so we concluded that only the intact OMVs could mediate the transfer of *bla*_CTX-M-55_. The *bla*_CTX-M-55_ transfer frequency of the control group (4.17 ± 2.13 × 10^−6^ CFU/mL) was similar to the Amoxicillin group (4.26 ± 2.47 × 10^−6^ CFU/mL), but the transfer frequency of the Enrofloxacin group was much higher (6.8 ± 2.14 × 10^−5^ CFU/mL) (Appendix A). Five colonies were randomly selected from each of the three groups for the detection of *bla*_CTX-M-55_, and it was found that all of them were *bla*_CTX-M-55_ positive *E. coli* C600(rif^r^) (Figure 4c). The antibiotic susceptibility testing of the pCTX-M-55 positive *E. coli* C600(rif^r^) indicates a resistance of ultra-broad spectrum β-lactam, FFC, and ENR (Appendix A). For a more complete understanding of the result, the whole genome of the transformants was sequenced (GenBank accession number JALGQP000000000), and the resistant genes and plasmid types were analyzed by Resfinder and Plasmidfinder of the CGE website. The results showed that the resistance genes of the recipient *E. coli* C600(rif^r^) including the chloramphenicol resistance gene *floR*, the β-lactamase genes *bla*_CTX-M-55_, *bla*_TEM-1B_, and the aminoglycoside resistance gene *aac(3)-IV*, which could not only mediate the resistance of aminoglycoside but also resistant to quinolones, which demonstrated that the OMVs could contain more than one resistance genes and mediate the transfer of these genes to the recipient. The plasmid of the recipient *E. coli* C600(rif^r^) including IncI2, IncI, Col440I, indicated that other resistance genes might also be located on these plasmids and transferred to *E. coli* C600(rif^r^). However, how these plasmids were enwrapped into OMVs still needs to be further studied.

## 3. Discussion

Horizontal gene transfer can promote the information exchange between bacteria, which is also an important means of bacterial evolution. The reason why *bla*_CTX-M-55_ is so widely spread in the *Enterobacteriaceae* is still unclear. There may be other ways that are easier for the horizontal transfer of *bla*_CTX-M-55_. Some studies have confirmed that Gram-negative bacteria secrete many phospholipids bilayer membrane structures named OMVs under natural living conditions, and produce more OMVs under physical, chemical, or stress conditions, indicating that the production of OMVs plays an important role in the survival of bacteria [25,26]. OMVs-mediated horizontal gene transfer is mainly carried out by wrapping nucleic acids, which avoids the hydrolysis of the internal nucleic acids by hydrolases, while also realizing direct delivery over long distances.

In this study, the genetic environment of *bla*_CTX-M-55_ carried by ESBL-producing APEC isolated from a sick-laying hen and OMVs-mediated gene horizontal transmission were analyzed. Our experimental results showed that the IncI2 plasmid bearing the *bla*_CTX-M-55_ could transfer to the *E. coli* C600 with the help of OMVs, which expanded the understanding of the transmission mechanism of *bla*_CTX-M-55_. In addition, we found that there were also IncI, Col440I inside the *E. coli* C600, which indicated that OMVs were effective tools to mediate the transfer of many plasmids, although how these plasmids came across the inner membrane and were enwrapped into OMVs is still unclear. Besides, our experiments demonstrated that different antibiotic treatments could lead to significant differences in the diameter, number, nucleic acid, and protein concentration of OMVs. The mean diameters of OMVs generated from Amoxicillin and Enrofloxacin treatment were 60.14 nm and 64.18 nm while the original was 79.42 nm. The concentration of OMVs generated after Amoxicillin and Enrofloxacin treatment was increased significantly (8.89 ± 0.36 × 10^11^ and 5.66 ± 1.2 × 10^12^) compared with the control group (2.26 ± 0.78 × 10^10^), which indicated that the generation mechanism of OMVs is varied by the antibiotic. 

Amoxicillin is a broad spectrum of β-lactam antibiotics, that mainly inhibits the synthesis of the bacterial cell wall to exert an antibacterial effect. Sharmin et al. found that OMVs originating from *nlpI* knockout *E. coli* would pack more plasmids than wild-type and *tolA* and *rseA* knockout hypervesiculating mutants, while increased membrane permeability due to peptidoglycan degradation would wrap more DNA entering OMVs [27]. These results explained the formation of OMVs and the cause of DNA entry into OMVs after Amoxicillin treatment. Although more OMVs were produced after Amoxicillin treatment, the gene transfer frequency was not significantly different from the untreated group, suggesting that the OMVs produced by APEC after Amoxicillin treatment may have other unknown functions. It has been known that the SOS response could stimulate vesicles formation. Federica et al. found that DNA damage-induced SOS response stimulates EVs formation in lysogenic *S. aureus* [28]. Enrofloxacin, as a quinolone antibiotic, mainly plays an antibacterial role by damaging the DNA, and therefore causing an SOS response. However, the mechanism of DNA entry into OMVs remains to be further elucidated. At the same time, we also found that bacterial OMVs secretion should be a complex mechanism involving the interaction of environment, microorganism, and microbial genetic evolution. A more complete exploration of the secretion mechanism would contribute to a deeper understanding of the one-health frame.

Collectively, our results suggested that the ESBL-producing APEC transferred *bla*_CTX-M-55_ into *E. coli* C600(rif^r^) via OMVs. The transfer frequency of OMVs could become higher when the APEC was treated with antibiotics, especially Enrofloxacin. Enrofloxacin, as a widely used veterinary antibiotic, according to the evidence of which in promoting resistance [29], should be used with more caution. Bacterial OMVs production is a continuous process, thus the delivery of drug resistance genes by bacterial OMVs will be efficient means. This phenomenon is consistent with previous reports on *Klebsiella pneumoniae* [22], indicating that it is likely to be a relatively common phenomenon in *Enterobacteriaceae*. Of course, more studies are required to further confirm this phenomenon. Our study confirmed that OMVs-mediated horizontal transfer of drug-resistant genes also existed in APEC, providing further evidence that OMVs may be efficient and universal tools for gene horizontal transfer, which also provides a scientific basis for rational use of antibiotics in aquaculture.

## 4. Materials and Methods

### 4.1. Strain Isolation and Antibiotic Susceptibility Testing

All strains used in this study were stored in our laboratory, including *E. coli* SCAO22 (isolated from the sick-laying hen sample), and *E. coli* C600 (rif^r^)strain, which is an engineering bacterium derived from *E. coli* K12 with high rifampicin resistance but is sensitive to other antibiotics (PMID: 17247495, 33603719, 32155266). Firstly, the sick-laying hen samples were cultured in BHI medium containing Cefotaxime sodium (4 μg/mL) for preliminary screening. Overnight cultures were streaked on EMB plates containing Cefotaxime sodium (4 μg/mL) for 16 h at 37 °C. Several single colonies with metallic sheen were selected and cultured in LB medium for a further 6 h. The genome was extracted and 16S rRNA sequencing was performed to further confirm *E. coli*. The minimal inhibitory concentrations (MIC) of Cefotaxime (CTX), Ceftazidime (CAZ), Amoxicillin (AML), Ampicillin (AMP), Meropenem (MEM), Aztreonam (ATM), Cefoxitin (FOX), Florfenicol (FFC), and Enrofloxacin (ENR) were determined by broth dilution method according to the CLSI guidelines. The clinical cut-off values of EUCAST were used for the interpretation of the MIC results. *E. coli* ATCC 25922 was used as a quality control strain for antibiotic susceptibility testing.

### 4.2. Whole-Genome Sequencing and Sequence Analyses

The genomic DNA of *E. coli* SCAO22 and *E. coli* C600(rif^r^) recipient strain that contained *bla*_CTX-M-55_ were extracted by TIANamp Bacteria DNA Kit (TIANGEN Biotech (Beijing) Co., Ltd., Beijing, China) according to the manufacturer’s recommendation, and then the gDNA were subjected to short-read WGS. Whole genomes were sequenced using the Illumina Hiseq platform (150-bp paired-end reads with about 200-fold average coverage). The clean data were assembled to draft genomes using the software SPAdes_3.13.0. Using different CGE tools to assign virulence genes and serotypes, for the genetic basis of AMR and for the detection of plasmid replicons of all contigs (https://cge.cbs.dtu.dk/services/ (accessed on 5 July 2021)) [30,31,32]. Based on the whole-genome splicing, the complete genome sequence of the plasmid bearing the *bla*_CTX-M-55_ was spliced by NCBI-BLAST and PCR sequencing. Using the online prokaryotic genome annotation website RAST (http://rast.nmpdr.org/ (accessed on 5 July 2021)) to annotate the full length of the spliced plasmid online, the complete plasmid was drawn by BRIG software.

### 4.3. OMVs Purification

*E. coli* SCAO22 was cultured in M9 medium at 37 °C, 220 rpm for 18 h and then used for isolating OMVs. Besides, 1/4 minimum inhibitory concentration of Amoxicillin (128 μg/mL) or Enrofloxacin (4 μg/mL) was used for the treatment of *E. coli* SCAO22, and then the culture was used for OMVs isolating. In brief, the bacterial culture was centrifuged at 4 °C 10,000× *g* for 20 min twice to further remove the bacterial cells, and the supernatant was retained. The supernatant was then filtered by 0.45 μM and 0.22 μM Millipore filters, respectively, to remove bacteria entirely. The filtrate was then concentrated with a 100 kDa Millipore ultrafiltration tube, and finally centrifuged at 4 °C 150,000× *g* for 2 h using the Beckman optimization L-80XP ultracentrifuge (SW 32 Ti rotor). After centrifugation, the supernatant was removed, and the precipitate was resuspended with PBS. The ultracentrifuge was repeated to obtain the OMVs crude extract. The crude extract of OMVs was slowly added to the pre-assembled size exclusion column (SEC), and then sterile PBS was slowly added. The 4–7th mL effluents were collected and concentrated in the 10 kDa Millipore ultrafiltration tube. The pure OMVs samples were stored at 4 °C for a short time and at −80 °C for a long time.

The obtained OMVs solution was filtered by 0.22 μM Millipore microfiltration membrane and treated with proteinase K (100 μg/mL) to digest any phage coats if exist. Then, 100 μL of the filtrate was taken and spread on an LB plate, and cultured for 24 h to confirm that there was no bacterial contamination. The protein concentration of OMVs was determined by PierceTM BCA Protein Assay Kit (Thermo Scientific TM, Waltham, MA, USA). DNase I (NEB, Ipswich, MA, USA) was added to degrade the nucleic acid outside of OMVs, which was inactivated at 75 °C for 10 min. The intact OMVs were lysed with 0.125% Triton X-100 and then the DNA was recovered with Monarch PCR & DNA purification kit. The nucleic acid concentration was measured according to Quant-iT TM PicoGreen^®^ dsDNA Reagent and Kits (Invitrogen, Waltham, MA, USA). Calculations were performed with Student’s *t*-test to calculate the difference of nucleic acid and protein concentration of OMVs purified from *E. coli* SCAO22 treated with different antibiotics.

### 4.4. Transmission Electron Microscopy

For TEM analysis, the OMVs sample was absorbed to glow-discharged copper grids with carbon-coated Formvar film and negatively stained with 2.0% (*w/v*) phosphotungstic acid at room temperature for 1–2 min. The samples were observed under transmission electron microscopy (TEM-1400PLUS, Japan) operated at 120 kV.

### 4.5. OMVs Size and Quantity Characterization by Flow Nanoanalyzer (nFCM)

For nFCM analysis, the sample stream is completely illuminated within the central region of the focused laser beam, and the detection efficiency is approximately 100%, which leads to accurate particle concentration measurement via single-particle enumeration. The concentration of each OMVs sample was determined by employing 100 nm orange FluoSpheres of known particles concentration to calibrate the sample flow rate and were recorded for 1 min. The flow rate and side scattering intensity were converted into corresponding particle concentrations and sizes according to the calibration curve [33,34].

### 4.6. Detection of Outer Membrane Protein A

To verify the presence of OmpA protein (an OMVs marker of *E. coli*) in the OMVs, the purified OMVs samples were separated in 12% polyacrylamide gel using tris-glycine running buffer, then transferred onto PVDF membrane. OmpA protein was detected using rabbit polyclonal against OmpA (Abmart, Shanghai, China). The goat anti-rabbit (Thermo Fisher Scientific, Waltham, MA, USA) secondary antibodies coupled with HRP were used as secondary antibodies. The specific bands were detected by chemoluminescence using BeyoECL Plus (Beyotime, Haimen, China) according to the manufacturer’s instructions. Proteins from *E. coli* lysate were used as the positive control.

### 4.7. β-NADH Oxidase Activity

To detect the inner membrane β-NADH oxidase activity, the β-NADH oxidase activity in OMVs was determined according to a previous study [21]. According to the instructions of NAD+/NADH detection kit (Beyotime, China), the purified OMVs from *E. coli* SCAO22 treated with different antibiotics were tested. Briefly, 20 μL of the pyrolysis OMVs sample was taken into the 96-well plate (with three repeat wells for every tested sample), 20 μL of NAD+/NADH extract solution and 90 μL of ethanol dehydrogenase working solution was added, incubated in dark at 37 °C for 10 min, and 10 μL of the chromogenic solution was added to mix well and incubated in dark for 30 min. The concentration of NAD+/NADH was measured at the absorbance of 450 nm.

### 4.8. Detection of bla_CTX-M-55_

The *bla*_CTX-M-55_ resistance gene in OMVs purified from *E. coli* SCAO22 treated with different antibiotics was detected. The nucleic acid extracted from OMVs in the above step was used as templates. Firstly, the resistance gene *bla*_CTX-M-55_ was detected by PCR, and the *E. coli* SCAO22 bacterial lysate was used as a positive control. The qPCR was performed with primer Q*bla*_CTX-M-55_F/Q*bla*_CTX-M-55_R (listed in Table 1) to calculate the copy numbers of *bla*_CTX-M-55_ in OMVs purified from *E. coli* SCAO22 treated with different antibiotics.

### 4.9. OMVs-Mediated Transfer Experiment of bla_CTX-M-55_

The experiment of *bla*_CTX-M-55_ gene transfer mediated by OMVs was carried out according to previous studies [21,22,23], with slight modifications. Briefly, different groups of OMVs were diluted to the same concentration (1 × 10^9^ particles/mL), 100 μL fresh *E. coli* C600(rif^r^) as the recipient (~10^7^ CFU) was mixed with 100 μL OMVs isolated from different SCAO22 (grown without antibiotics (the control group) or with 128 μg/mL Amoxicillin (the Amoxicillin group) or 4 μg/mL Enrofloxacin (the Enrofloxacin group)), respectively. The mixture was pretreated with DNase I (degrade the external DNA to rule out bacterial DNA transformation) and protease K (degrade the phage coats to rule out transduction) and stood for 4 h at 37 °C, and then the mixture was supplemented with an 800 μL LB medium and cultured at 37 °C 220 rpm for 4 h. After that, the mixture was supplemented with an 8 mL LB medium and further cultured for 24 h. The cultured bacterial was centrifuged and the precipitate was resuspended with 1 mL PBS, and 100 μL bacterial liquid was spread on a plate with rifampin (200 μg/mL) or a plate with rifampin (200 μg/mL) together with Cefotaxime (4 μg/mL), respectively, and cultured for 24 h, the transfer frequency was calculated by colony counting. To rule out bacterial DNA transformation as being responsible for the transfer of antibiotic resistance to *E**. coli* C600, the plasmid (10 ng) extracted from SCAO22 was incubated with *E. coli* C600 and cultured in the same method. Besides, to test whether the integrity of OMVs has an impact on this transfer mode, an equal amount of OMVs lysed with 0.125% Triton X-100 was incubated with *E. coli* C600(rif^r^), the mixture was pretreated with DNase I and protease K, and cultured in the same method. Finally, PBS was incubated with *E. coli* C600(rif^r^) as the negative control.

## Figures and Tables

**Figure 1 pathogens-11-00481-f001:**
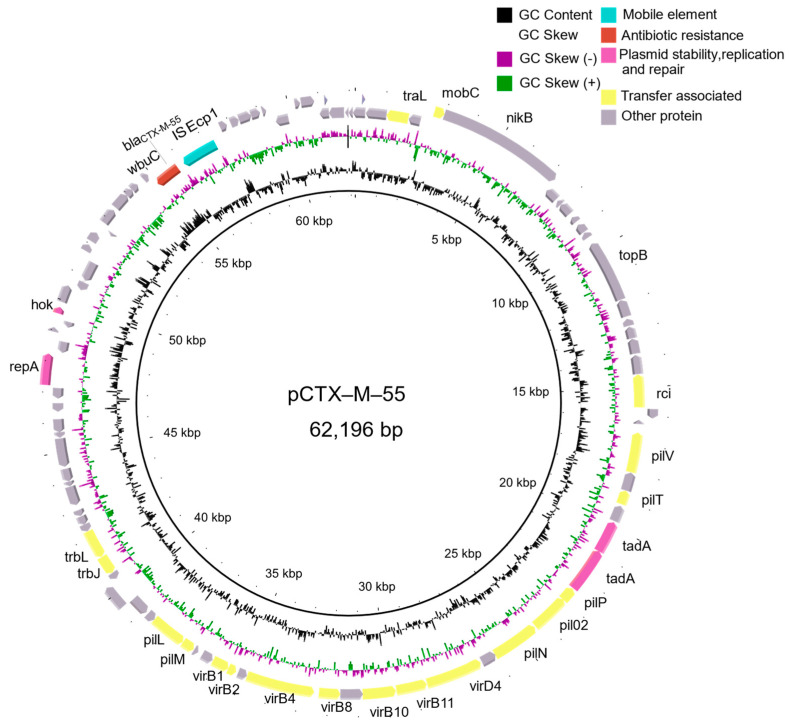
Characteristics of the *bla*_CTX-M-55_ bearing IncI2 plasmid identified in this study (GenBank accession number OL539428). The red box shows the target gene discussed in this study. Genes with plasmid stability and transfer functions as well as mobile elements are shown in pink, yellow, and aquamarine, respectively. Other genes are displayed in gray.

**Figure 2 pathogens-11-00481-f002:**
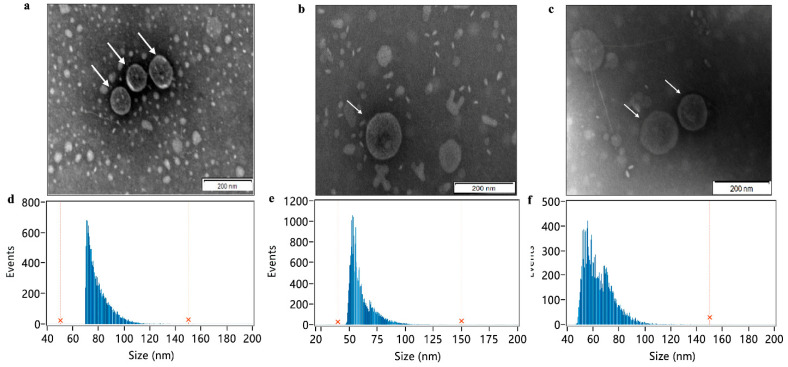
Evaluation of the extracted OMVs. (**a**–**c**) TEM analysis of the extracted OMVs from the control group, Amoxicillin treatment group (128 μg/mL), and Enrofloxacin treatment group (4 μg/mL) (scale bar: 200 nm). (**d**–**f**) nFCM measurement of the particle size (diameter) distribution of OMVs from the control group, Amoxicillin treatment group (128 μg/mL), and Enrofloxacin treatment group (4 μg/mL).

**Figure 3 pathogens-11-00481-f003:**
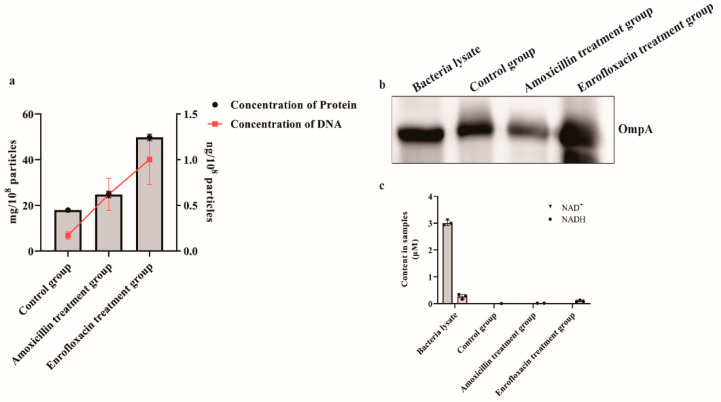
Characteristic analysis of OMVs in different groups. (**a**) The nucleic acid and protein concentration of OMVs in different groups. (**b**) Western blot analysis of OmpA. (**c**) Quantification of inner membrane-associated β-NADH oxidase activity. The bacterial cell lysate was a positive control. No β-NADH oxidase activity was detected in OMVs released from the control group, 128 μg/mL AML treatment group, and 4 μg/mL ENR treatment group. Error bars indicate standard deviations.

**Figure 4 pathogens-11-00481-f004:**
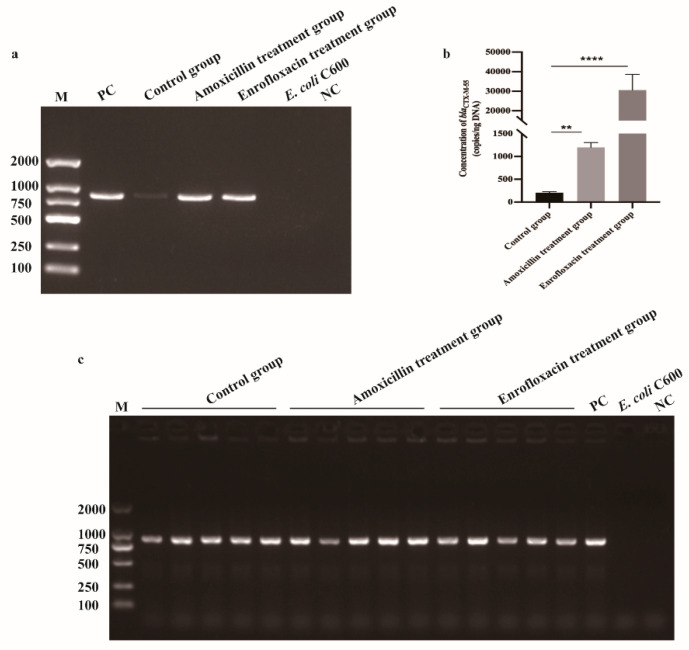
Detection of *bla*_CTX-M-55_. (**a**) Detection of *bla*_CTX-M-55_ in OMVs by PCR. The nucleic acid extracted from three groups of OMVs isolated from different treatment SCAO22 (control group, 128 μg/mL Amoxicillin treatment group, and 4 μg/mL Enrofloxacin treatment group), were used as templates. Genomic DNA of *E. coli* SCAO22 was used as the positive control, genomic DNA of *E. coli* C600(rif^r^) and sterile water were used as the negative control. (**b**) Quantification of *bla*_CTX-M-55_ in three groups of OMVs (control group, 128 μg/mL Amoxicillin treatment group, and 4 μg/mL Enrofloxacin treatment group) by qPCR. The copy number of *bla*_CTX-M-55_ was calculated according to the standard curve: y = −3.478x + 36.570, R^2^ = 0.997. (**c**) Colony-PCR detection of *bla*_CTX-M-55_ from three groups transformants (*E. coli* C600 co-incubation with OMVs-control, OMVs-128 μg/mL Amoxicillin treatment, OMVs-4 μg/mL Enrofloxacin treatment). Five colonies were randomly selected from each of the three groups for the detection of *bla*_CTX-M-55_, control group (Line 1–5), Amoxicillin treatment group (Line 6–10), and Enrofloxacin treatment group (Line 11–15). Genomic DNA of *E. coli* SCAO22 was used as the positive control (Line 16), *E. coli* C600(rif^r^) (Line 17), and sterile water (Line 18) were used as the negative control. ** *p* < 0.01, and **** *p* < 0.0001.

**Table 1 pathogens-11-00481-t001:** The primers used in this study.

Primer	Sequence (5′-3′)	Size (bp)
ctx-m-55f	ATGGTTAAAAAATCACTGCGCCAGT	25
ctx-m-55r	TTACAAACCGTCGGTGACGAT	21
Q*bla*_ctx-m-55_F	AACCGTCACGCTGTTGTTAGGAAG	24
Q*bla*_ctx-m-55_R	AATCAATGCCACACCCAGTCTGC	23

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
