# Peer review of "Outer Membrane Vesicles of Avian PathogenicEscherichia coli Mediate the Horizontal Transmission of blaCTX-M-55"

_pathogens, 2022, doi:10.3390/pathogens11040481_

Round 1
Reviewer 1 Report
Presented manuscript is focused on OMV structures, which seems to play significant role in bacterial virulence. After reading the whole work I have a few suggestions. Introduction is too long, I recommend to shorten this part and include the most important facts. Moreover, please add the explanation why OmpA is important in such type of investigations. Lines 108 and 117 contradict each other, once authors say that OMV hgt is universal, and then that horizontal gene transfer involving OMVs is not universal, please adopt a common version. Correction of the language is recommended. There are somewhere tiny mistakes.
Author Response
Response: Thank you very much for your kind help in the evaluation of our manuscript, and we revised it in the manuscript. We have shortened the introduction. And since OMVs extraction is a challenging job, pure bacterial OMVs are required for better interpretation of OMVs biology. The impurity of extracted vesicles will adversely affect the experimental result. So, it is necessary to assess the quality of OMVs. Outer membrane proteins may have utility as technical markers for assessing the purity of E. coli OMVs preparations. According to the reference of PMID: 31275533, outer membrane protein A was highly enriched in purified OMVs, which could serve as a ubiquitous protein marker of E. coli OMVs just as Alix and CD63 are used as common markers for eukaryotic exosomes. So, it is necessary to detect ompA.
Reviewer 2 Report
General
Li et al. demonstrated that transferring blaCTX-M-55 from an avian Escherichia coli strain (SCAO22) to the recipient E. coli C600 strain could be achieved by outer membrane vesicles (OMVs) of SCAO22 strain. This manuscript was well written, the experiments were well designed and carried out, and the results supported their hypothesis.
Have the authors come across any literature indicating that APEC could transfer bla genes by conjugation test? For example, can the SCAO22 strain mix with E. coli J53 (ATCC BAA-2730), sodium azide resistant, and examine if the recipient E. coli J53 would be CTX-M-15 positive and sodium azide resistant? (Tamang et al., Antimicrob. Agents Chemother. 2012, 56, 2705–2712)
The discussion presented in this manuscript was kind of weak. I suggest the authors go through the whole section, remove any general background information, or move to the introduction section. Discussion should focus on the results of this manuscript.
Specific comments
Results
Line 143 to 144: Please rewrite this sentence; it has a grammatical error.
Table S1: I suggest the breakpoints of the antibiotics listed in the table should be indicated or as a footnote with the table. In addition, you should mention Table S1 somewhere in the results section line 132 to line 144.
Discussion
Line 219-259: Please go through these three paragraphs again. Some of them should be placed in the Introduction section. It would help if you focus on discussing the results generated from the present study.
References: Please correct the No. 1 reference; need detailed information. Reference No. 21: Only the first letter of the title needs to be capitalized.
Reviewer 3 Report
Summary
The authors isolated a multi-drug resistant avian E. coli and sequenced the genome. The strain was found to have multiple antibiotic resistance genes and had a conjugative plasmid carrying a beta lactamase gene, blaCTX-M-55. They purified outer membrane vesicles (OMVs) and asked whether presumed DNA inside those vesicles could mediate transfer of this blaCTX-M-55 to a susceptible strain. Remarkably, the answer was yes- OMVs can mediate horizontal transfer of DNA inside. While this is not a new finding per se, it adds to the growing body of literature in different organisms that OMVs are a new mechanism of genetic exchange.
Major suggestions
- To add a little more rigor, I suggest additional controls be conducted. The authors demonstrate OMVs can mediate the transfer of DNA to confer antibiotic resistance to a susceptible isolate. What happens if the OMVs are pre-treated with a detergent to solubilize the OMVs and release the DNA? Does the transfer still happen, now presumably by transformation? And same experiment of detergent but plus DNAse, which should then destroy the DNA and abolish the transfer. Something needs to be shown that the DNase used is actually working under the experimental conditions and buffer the OMVs are stored in, because this is a crucial reagent used on the OMVs to basically rule out DNA being on the outside of OMVs and mediating transfer by transformation. No control is provided that the DNAse is working- it is just assumed.
- The discussion is overly long and speculative at times. I recommend cutting the discussion by 25%.
- Line208. Provide GenBank Accession # for recipient C600 following the transfer and expand on the genomic discussion of the strain before and after the transfer- is the plasmid the ONLY difference, or are there other changes (do other antibiotic resistance genes and other genes from SCA022 end up in C600)? Also it needs to be very clear that the resulting antibiotic resistant recipients are distinct in genomics sequence from the parent SCA022- they are truly C600 lineage. [My thinking: OMVs could transfer other genes, not just bla, so the recipient could have many additional changes, which would be interesting; also some mild concern that the SCA022 strain just gained rifampin resistance and the “recipients” are not C600 at all, but are SCA022 descendants- or some unusual hybrid of the two organisms].
- Why did the authors not also check the MICs of sulfonamides, tetracycline and kanamycin/gentamicin of both donor (whose genomic sequence is reported here to contain potential antibiotic resistance loci) and C600 recipient before and after OMV transfer? Such data would provide more complete picture as the recipient C600 before transfer should still be susceptible to the other antibiotics, and should remain susceptible to these antibiotics after OMV transfer, unless there is OMV transfer of other resistance traits.
- Line 213. Table S2 is actually Table S1 in the supplement. I do not understand the data in this table fully. The recipient C600 before transfer is susceptible to all the antibiotics, as expected. But why did this strain gain increased MIC (increased resistance) to antibiotic classes that have nothing to do with bla transfer (ENR, FFC)? The recipient, after transfer, has a resistance profile that is nearly identical to the SCA022 strain, except MEM. The data raises the possibility of the recipients actually just being descendants of SCA022, not C600 lineage (see point 3 above). The genomics comparisons of donor versus recipient and recipient after transfer needs to be discussed more fully- would add more rigor to the study.
Minor suggestions
- English/sentence structures. There are too many examples to mention, but the manuscript needs to be polished for sentence structure and clarity in English language. A few examples just in the abstract: Line 16- add “on the” after “clarified”; line 23- add “was” before “located”; line 28- sentence is unclear.
- Line 72- provide reference [ref 4] for the PMID #. Sentence unclear “the spread scope”-?
- Lines 77-83. Gram (+) bacteria have no outer membrane and thus do not produce outer membrane vesicles, so this section is inaccurate and misleading. They are just referred to as extracellular vesicles (EVs) in Gram (+)- clarify this whole section. LPS would only be found in OMVs, not EVs.
- Lines 134-136, 153, 164. All abbreviations used in the manuscript must be spelled out on the first usage- this is the first time all these names come up (CTX, CAZ,…; nFCM; WB).
- Line 133. Please provide GenBank accession # for the whole genomic sequence data for the avian E. coli strain.
- Fig. 1. The text within the figure (gene names) and the legend is too small and pixelated. Please enlarge text and re-draw figure so that pixelation is minimized.
- Fig. 2 legend. Provide concentrations of antibiotic used in these experiments.
- Fig 4 legend. Please explain the experiment better; it is unclear, especially in C what is being investigated- is C data on the C600 recipient of OMVs coming from the avian strain grown with or without antibiotics?
- Provide reference or source and genotype for E. coli C600, and what gene is conferring the rifampin resistance (rpoB?).
- Lines 204 and 205- check the numbers (4.26 +/-2.47 are the same in both lines)- they are identical yet they seem to be two different groups.
- Line 276-278. The authors speculate and then say this “well explains”… but did not carry out experiments to investigate the SOS response. Suggest softening the wording or remove.
- Lines 285-293. Delete or eliminate most of this. Nothing was done on ABC transporters or Type 6 secretion and this is the first time these topics are introduced; this section really does not add to the discussion, but dilutes its focus and impact. Lines 294-307 might be all that is needed.
- Line 302. Provide ref # for the PMID #.
- Reference #1 seems to be messed up or incomplete citation; italicize all genus and species names and any gene names in the titles within all references.
Round 2
Reviewer 3 Report
Summary
The authors have responded favorably to most of the previous comments on the manuscript. There are minor editorial changes to further suggest and still lingering concerns over the response to prior major point #1.
Major suggested changes
- Reply on previous major suggestion #1. The authors get close to addressing the concern, but it is still incomplete. Focus only on OMVs from SCA022 grown without antibiotics for this experiment and the rif resistant C600 recipient and score for rif/cef. The proper experimental setup should be OMVs (intact; positive control), OMVs + TX-100 (no DNAse) [this one might still mediate transfer because DNA would presumably be present- might even lead to higher recovery of resistant colonies, or may not work at all if E. coli recipient does not take up DNA naturally under the experimental conditions, which actually works in the author’s favor], OMVs +TX-100 + DNAse [should abolish all DNA transfer], and OMVS + DNase (no Triton-X-100; removes any exogenous DNA that might be sticking to the outside of intact OMVs; the DNase should not cross the OMV membrane). Same concentration of OMVs needs to be in all groups. I’m not clear why proteinase K is needed in these experiments as it is irrelevant to ruling out DNA transformation as the mechanism. The experiment they described in the rebuttal and revised manuscript is incomplete and still leaves room for DNA transformation. Given the huge (and exciting) importance of the finding of OMV-mediated DNA transfer of antibiotic resistance genes, there needs to be sufficient rigor so that the scientific community will better accept the validity of the finding. I do believe the authors have discovered something great here, but I have concern the scientific community may question it if the authors have not done the rigorous experiment. Having the rigor may lead to higher citation of the published manuscript.
Minor suggested changes
- Lines 44-46. This is a poorly worded/structured sentence, with run-on clauses. “Preferentially hydrolyzes cefotaxime” is better wording. Re-write the sentence.
- Line 50. Mean to say? “have so far been identified in clinical pathogens”. If more than 1 pathogen other than E. coli, do they mean other members of Enterobacteriaceae or is it found broader than this Gram negative family? Or do they only mean the clinical E. coli pathogens?
- Fig. 1. The figure is improved but only marginally. The entire figure should be increased in size, along with the accompanying text within the figure, as the pixelation and font sizes remain a problem and the text in the figure is still difficult to read.
- Line 195. Change to “mediate”
- Line 274. Provide the published reference for strain C600 or the rif resistant derivative.
- Line 19. Change “verify” to “test”. Scientists test hypotheses in an unbiased way. To seek to verify a hypothesis can appear to be biasing the experiments to only demonstrate the hypothesis is correct and favorable to what the authors hope to find.
- Lines 27-29. The last sentence should switch back to present tense (change “contributed” to “contribute” and “supported” to “support”. I think the last clause about restricting enrofloxacin use in the poultry industry may be too strong and go too far as the paper is more about OMV-mediated beta lactamase (bla) transfer than fluoroquinolone resistance.
- Line 187-189. Reword: A rifampin resistance (MIC >1024 ug/ml) derivative of E. coli C600 was used as the recipient strain and incubated with OMVs isolated from SCA022 grown without antibiotics or with Amoxicillin (128 ug/ml) or with Enrofloxacin (4 ug/ml). To first rule out bacterial DNA transformation as being responsible for the transfer of antibiotic resistance to C600, OMVs were incubated with Triton-X-100 (0.125%; lyses OMVs), DNase (degrades DNA) and proteinase K (degrades protein) and compared with intact OMVs. Only intact OMVs led to cefotaxime-resistant colony recovery (data not shown). ……. Lines 189-192: This sentence is not clear either/has English issues.
Author Response
Please see the attachment

This manuscript is a resubmission of an earlier submission. The following is a list of the peer review reports and author responses from that submission.